# Tissue-specific and *cis*-regulatory changes underlie parallel, adaptive gene expression evolution in house mice

Sylvia M. Durkin[1]*, Mallory A. Ballinger[1¤], Michael W. Nachman[1]*

1 Museum of Vertebrate Zoology and Department of Integrative Biology, University of California, Berkeley, Berkeley, California, United States of America

¤ Current address: Department of Biology, Utah State University, Logan, Utah, United States of America
* sdurkin@berkeley.edu (SMD); mnachman@berkeley.edu (MWN)

**Data Availability Statement:** All sequencing reads generated in this study are available on the NCBI Sequence Read Archive under accession

## Abstract

Changes in gene regulation have long been appreciated as a driving force of adaptive evolution, however the relative contributions of *cis*- and *trans*-acting changes to gene regulation over short evolutionary timescales remain unclear. Instances of recent, parallel phenotypic evolution provide an opportunity to assess whether parallel patterns are seen at the level of gene expression, and to assess the relative contribution of *cis*- and *trans*- changes to gene regulation in the early stages of divergence. Here, we studied gene expression in liver and brown adipose tissue in two wild-derived strains of house mice that independently adapted to cold, northern environments, and we compared them to a strain of house mice from a warm, tropical environment. To investigate gene regulatory evolution, we studied expression in parents and allele-specific expression in F1 hybrids of crosses between warm-adapted and cold-adapted strains. First, we found that the different cold-adapted mice showed both unique and shared changes in expression, but that the proportion of shared changes (i.e. parallelism) was greater than expected by chance. Second, we discovered that expression evolution occurred largely at tissue-specific and *cis*-regulated genes, and that these genes were over-represented in parallel cases of evolution. Finally, we integrated the expression data with scans for selection in natural populations and found substantial parallelism in the two northern populations for genes under selection. Furthermore, selection outliers were associated with *cis*-regulated genes more than expected by chance; *cis*-regulated genes under selection influenced phenotypes such as body size, immune functioning, and activity level. These results demonstrate that parallel patterns of gene expression in mice that have independently adapted to cold environments are driven largely by tissue-specific and *cis*-regulatory changes, providing insight into the mechanisms of adaptive gene regulatory evolution at the earliest stages of divergence.

## Author summary

The parallel movement of organisms into novel environments provides an opportunity to understand the molecular basis of adaptation and the repeatability of this process.

BioProject ID PRJNA1009445. All other data are within the paper and Supporting Information files.

**Funding:** Funding and support for this work was provided by the National Institutes of Health (R01 GM074245, R01 GM127468, and R35 GM149304 to M.W.N.). This work used the Extreme Science and Engineering Discovery Environment (XSEDE), which is supported by National Science Foundation grant number ACI-1548562 to M.W.N.. S.M.D. was supported by, and received salary from, a NIH NRSA T-32 training grant GM132022. M.A.B. was supported by, and received salary from, a National Science Foundation Graduate Research Fellowship (DGE-1106400), a Junea W. Kelly Museum of Vertebrate Zoology Graduate Fellowship, and a Philomathia Foundation Graduate Fellowship. The funders had no role in study design, data collection and analysis, decision to publish, or preparation of the manuscript.

**Competing interests:** The authors have declared that no competing interests exist.

Mutations affecting the expression of genes are known to underlie much of adaptive evolution. Such mutations can arise in *cis-* (near the gene of interest) or in *trans-* (at a distant locus), but the relative contribution of these different kinds of changes to adaptation is poorly understood, especially during very recent divergence. Here, we compared evolved gene expression differences between a warm-adapted house mouse strain and two different cold-adapted strains that have independently evolved similar phenotypic traits, such as increased body size and decreased extremity length during the last few hundred years. Using crosses between warm-adapted and cold-adapted mice, we found that mutations acting in a context specific manner (*cis*-regulatory and tissue-specific changes) predominate expression divergence and are more likely to be involved in parallel evolution. We used population level selection scans in wild animals to identify regions of the genome under selection and combined these findings with the gene expression data to identify candidate genes underlying adaptation to novel environments. Together, our work describes the gene regulatory dynamics of rapid environmental adaptation, and the repeatability of these patterns over multiple instances of adaptation.

## Introduction

Understanding the genetic architecture of adaptive evolution is a long-standing goal in evolutionary biology. Differences in gene regulation are known to underlie adaptation to new environments across a diverse set of organisms and phenotypes including armor plating and pelvic spine reduction in sticklebacks, coat color differences in deer mice, and the ability to digest lactose in humans [1–4].

Evolved changes in gene expression can occur via changes in *cis*-regulatory elements, which act locally in an allele-specific manner, or *trans*-regulatory elements, which act distally and can potentially affect many genes [1,5]. Since *cis*-regulatory changes can modulate gene expression in a tissue-specific and temporal-specific manner, these variants are expected to be less pleiotropic and may therefore evolve more easily [1,5]. In contrast, *trans*-acting changes may affect the expression of many genes and thus are expected to be more constrained. However, alterations to the wider regulatory environment via *trans*-acting changes may be favored during early stages of adaptation, when populations are further from their fitness optima [6,7].

Given that changes in gene expression underlie much of adaptive evolution, understanding the contributions of *cis-* and *trans*-changes to gene regulation will help shed light on the molecular mechanisms and genetic architecture of evolutionary change. Several studies have evaluated the contributions of *cis-* and *trans*-acting mutations to gene expression by studying allele-specific expression in hybrid individuals or by mapping expression quantitative trait loci (eQTL) (reviewed in [5,8]). One emerging pattern is that *cis-* variants often contribute more to expression divergence between species, while *trans-* variants dominate within-species expression variance [9–13]. This observation is usually attributed to the larger mutational target for *trans*-acting variants compared to *cis*-acting variants. Between species, *cis*-acting variants have sometimes been found to constitute a greater proportion of regulatory variants at increasing levels of evolutionary divergence [14]. Despite these patterns, some studies have found a larger role for *cis*-acting changes over short evolutionary timescales. For example, comparisons between subspecies of house mice have revealed a greater proportion of *cis*-regulatory changes compared to *trans*-regulatory changes in liver [15] and in testes [16]. Comparisons among lines of *Drosophila melanogaster* have also revealed a preponderance of *cis*-regulatory effects [17]. Different studies on the same species have sometimes reached different conclusions. For

example, in comparisons between marine and freshwater populations of stickleback fish, Hart et al. [18] found a greater role for *trans*-acting changes in governing expression differences in pharyngeal dental tissue, while Verta and Jones [19] found *cis*-acting changes to predominate expression differences in the gills. In sum, it remains unclear whether less pleiotropic changes (*cis*-regulated or tissue specific genes) or more pleiotropic changes (*trans*-regulated or broadly-expressed genes) are most likely to contribute to adaptive gene expression evolution over short time scales.

Recent, repeated colonization of novel environments not only provides an opportunity to assess the nature of evolutionary changes over short time scales, but also to study parallel evolution. When different populations have experienced similar environments and selection pressures, repeated patterns provide evidence that the observed changes are driven by selection. Situations in which phenotypes have evolved in parallel provide an opportunity to ask whether parallelism is similarly seen at the molecular level.

House mice (*Mus musculus domesticus*) represent a useful system in which to explore the gene regulatory basis of environmental adaptation and the degree of parallelism in expression divergence. House mice are found throughout inhabited regions of the world [20] and display morphological, physiological, and behavioral phenotypes reflective of their environment [21,22]. For example, mice from northern latitudes have larger bodies, shorter extremities, and build larger nests, conforming to well-documented ecogeographic patterns in mammals [21–23]. Shortened extremities and increased body size are viewed as thermoregulatory adaptations since they decrease the ratio of surface area to volume, thereby minimizing heat loss [24]. Importantly, these phenotypes have evolved repeatedly in house mice in response to cold temperature. The phylogeny of North American populations shows that house mice colonized northern environments at least twice, once in the west and once in the east, providing evidence of parallel evolution in North America [25]. However, this conclusion does not speak to whether the independent instances of selection in northern environments acted on new mutations, standing variation, or alleles introduced by rare, long-distance migrants. The absence of long-haplotypes, as expected with hard selective sweeps of new mutations, suggests that most selection acted on standing variation, either from ancestral polymorphism or from alleles introduced by migration [22,25], much like parallel instances of evolution in freshwater populations of threespine sticklebacks [26]. Finally, previous studies have shown that gene regulation is likely to be a primary mediator of adaptive evolution in house mice, as scans for selection have identified adaptive mutations primarily in non-coding regions of the genome [22].

Here, we analyze gene expression in inbred strains of house mice derived from two, independently-colonized northern populations and one equatorial population to address several key questions related to gene expression evolution, including the degree to which expression divergence is shared between parallel cases of adaption, whether this expression evolution is governed by primarily *cis*- or *trans*-acting changes, as well as whether those changes are largely tissue-specific or broadly expressed. We also integrate expression data with scans for selection in wild-caught mice to connect expression differences identified in lab-reared animals to selection acting in natural populations. We find strong evidence for parallel adaptive evolution due to changes in the expression of *cis*-regulated and tissue-specific genes in the early stages of adaptation to novel environments.

## Results

### Phenotypic divergence between cold-adapted and warm-adapted mice

Previous work based on phylogenetic analysis of house mouse populations across eastern and western North America showed that mice in upstate New York and mice in Western Canada

represent independent range expansions into northern habitats from southern locations and independent cases of adaptation to cold climates with parallel differences in body size [25]. We thus chose one strain from Saratoga Springs, New York, USA (SARA) and one strain from Edmonton, Alberta, Canada (EDMA), and compared these to one strain from Manaus, Brazil (MANA) (Fig 1A, data from WorldClim [27], available from: www.worldclim.com/version2). Lab-reared inbred strains of mice from these localities display divergent, environmentally adaptive phenotypes under common conditions, suggesting that they have a genetic basis (Fig 1C–1F). In particular, mice from New York and Alberta are larger and have shorter tails, ears, and feet when compared to mice from Manaus. These differences conform to Bergmann's Rule (larger size in cold environments) and Allen's Rule (shorter extremities in cold environments) and are thought to reflect thermoregulatory adaptations.

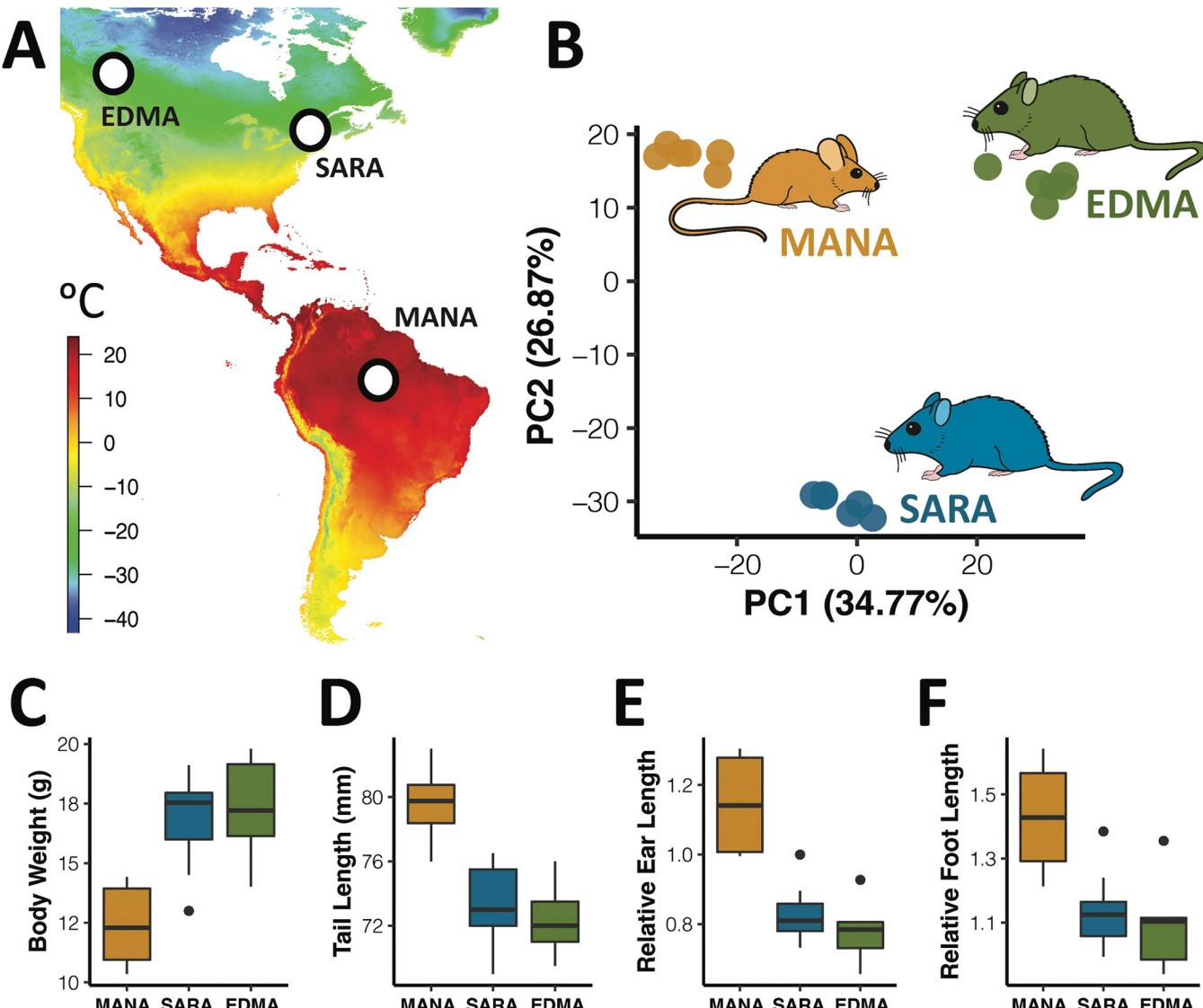

**Fig 1. Morphological and gene expression variation in focal mouse strains.** A). Variation in minimum annual temperature of original sampling localities for focal mouse strains. Data from WorldClim [27], available from: www.worldclim.com/version2. B). Principal component analyses for liver RNA-seq data separates samples based on original sampling locality. BAT data shown in S1 Fig. C-F). Morphological variation in male mice collected for RNA-seq reflects environmental adaptation to differing climates.

## Expression divergence between cold-adapted and warm-adapted mice

To study expression evolution between cold-adapted and warm-adapted mice, we generated RNA-seq data from the liver and brown adipose tissue (BAT) of six males each from the SARA, EDMA, and MANA strains. We chose liver and BAT, as liver underlies many metabolic processes involved in environmental adaptation in house mice [22,28], and BAT is involved in cold adaptation via its role in non-shivering thermogenesis [29,30]. Principal component (PC) analysis of expression separates samples primarily by tissue-type (S1A Fig), agreeing with studies investigating divergent gene expression across multiple tissues [31,32]. Within each tissue, PC analysis separates individuals by strain with PC1 and PC2 two explaining 34.8% and 26.9% of expression variation in liver, respectively (Fig 1B), and 30.3% and 27.9% of expression variation in BAT, respectively (S1B Fig). The differences in expression between strains undoubtedly reflects both strain-specific differences as well as population-level differences, although previous work has shown that expression variation in wild-derived inbred mice from different populations mostly reflects population-level differences [22].

To quantify the proportion of unique and shared changes associated with independent instances of adaptation to northern environments, we compared evolved gene expression changes between SARA and MANA to evolved changes between EDMA and MANA using DESeq2 [33]. For liver and BAT, respectively, we found 4,121 and 4,160 significantly differentially expressed genes (DEGs) between SARA and MANA, and 4,936 and 4,585 DEGs between EDMA and MANA (FDR < 5% for all differential expression analyses). Of these genes, 2,361 and 2,231 were shared between SARA and EDMA comparisons for liver and BAT, respectively (Figs 2A and S2A). Thus, roughly half of the DEGs were shared between populations and roughly half were unique to each population. Nonetheless, the number of shared genes is 1.68 and 1.72 times greater than expected by chance in liver and BAT, respectively, as measured through permutation tests (see methods for details, Figs 2B and S2B). Shared changes include those due to parallel evolution in New York and Alberta house mice, as well as those that occurred in the common ancestor of North American mice or along the lineage leading to Manaus, as discussed in the "*Signals of selection to northern environments*" section below.

To further explore these shared changes, we measured the correlation between SARA-MANA expression differences and EDMA-MANA expression differences. $\mathrm{Log}_2$ fold changes between SARA and MANA are highly correlated with those between EDMA and MANA (Fig 2C, Spearman's r = 0.54 for liver, S2C Fig, Spearman's r = 0.53 for BAT). Moreover, significant DEGs for both comparisons have larger correlation coefficients than the transcriptome as a whole (Figs 2C and S2C). Because spurious correlations can occur when comparing two ratios that share the same denominator, we reanalyzed the RNA-seq data using different subsets of the MANA mice to define differential expression in the two comparisons (i.e. EDMA mice were compared to MANA mice 1, 2, and 3, while SARA mice were compared to MANA mice 4, 5, and 6) and all significant correlations were recapitulated (S3 Fig).

To investigate the importance of these expression changes to phenotypic changes between cold-adapted and warm-adapted mice, we next asked whether genes involved in increased body size were implicated in parallel expression evolution to a greater degree than the transcriptome more generally, since larger body size is a trait that has evolved independently in New York and Alberta mice [25]. We defined a list of 412 "increased body size genes" using phenotypes defined by the mouse genome informatics (MGI) database (MP code: 0001264). Supporting this hypothesis, we found that expression divergence in increased body size genes was highly correlated between the comparisons in both BAT and liver (Fig 3, Spearman's r = 0.82 for liver, 0.69 for BAT), showing greater correlation coefficients than for either the transcriptome as a whole or for DEGs.

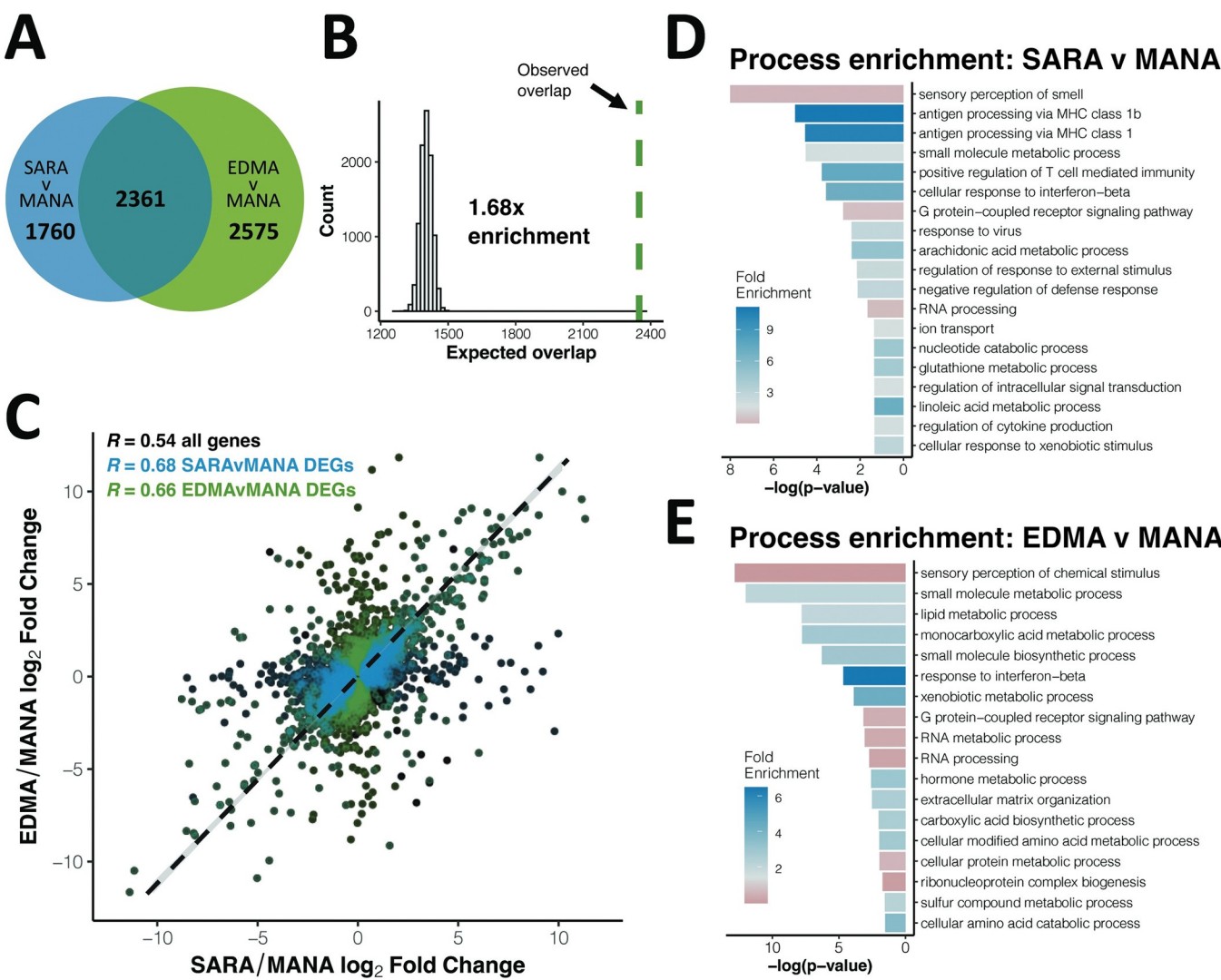

**Fig 2. Parallel gene expression evolution in liver.** A). Overlap of significantly differentially expressed genes (DEGs) between SARAvMANA and EDMAvMANA comparisons (FDR < 5%). B). Expected vs. observed overlap between SARAvMANA and EDMAvMANA DEGs. C). Correlation between SARAvMANA $\log_2$ fold change and EDMAvMANA $\log_2$ fold change for all genes (black), SARAvMANA DEGs (blue) and EDMAvMANA DEGs (green). Dashed line represents reduced major axis regression for all genes. Correlation coefficients are Spearman's R. D-E). Gene ontology enrichment for SARAvMANA DEGs (D) and EDMAvMANA DEGs (E).

## Enrichment of immune related functions among differentially expressed genes

To categorize the potential function of genes involved in gene expression evolution in cold-adapted mice, we used gene ontology analyses via PANTHER [34]. We found high enrichment for biological processes related to immune functioning for both the SARA and EDMA comparisons in both BAT and liver, such as antigen processing, response to interferon-beta, and regulation of T cells (Figs 2D, 2E, S2D and S2E). Additionally, we found enrichment for metabolism related processes across comparisons and tissues, including lipid metabolism, small molecule metabolism, and various other metabolic processes, which may be indicative of metabolic evolution related to cold adaptation at the transcriptomic level.

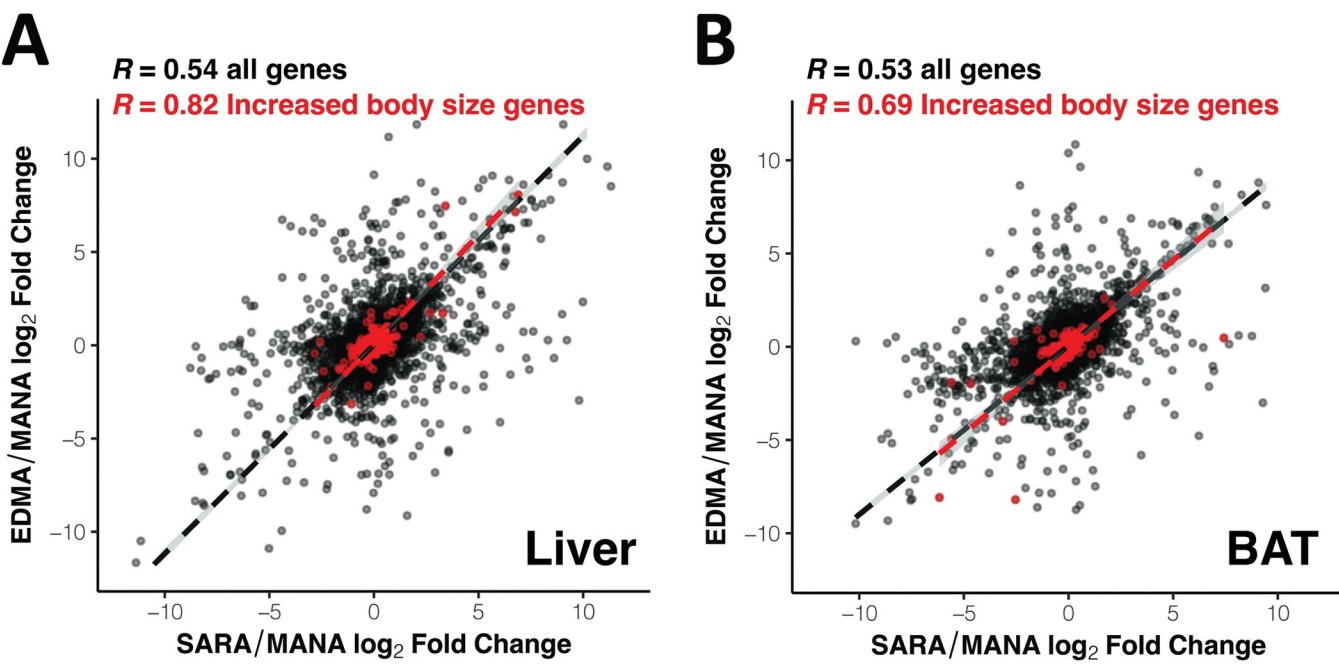

**Fig 3. Parallel expression evolution in increased body size genes.** Correlation between SARAvMANA $\log_2$ fold change and EDMAvMANA $\log_2$ fold change in liver (A) and BAT (B) for all genes (black) and increased body sized genes (red). Dashed lines represent reduced major axis regression. Correlation coefficients are Spearman's R.

## Expression divergence is primarily governed by *cis*-regulatory changes

We next asked whether gene expression differences are primarily governed by *cis*-regulatory changes or *trans*-regulatory changes by measuring allele-specific expression (ASE) in liver and BAT from F1 hybrids (both SARA X MANA and EDMA X MANA). Because F1 hybrids possess one allele from each parent within the same *trans*-acting environment, genes showing allele-specific differences in expression in F1s are governed by a *cis*-acting change [35].

For both comparisons and both tissues, we found that *cis*-acting changes underlie most expression divergence. While most gene expression was conserved between cold-adapted and warm-adapted mice, in the liver we found that 8.6% and 8.0% of all genes were regulated in *cis*- only, while 1.7% and 4.6% were regulated in *trans*- only for SARA vs. MANA and EDMA vs. MANA, respectively (Fig 4A–4B). Expression in BAT showed similar trends (S4A–S4B Fig). Moreover, we found a greater proportion of *cis*-regulated genes than *trans*-regulated genes when we looked at all DEGs, shared DEGs, or DEGs that were unique to the EDMA and SARA comparisons (see S1 Appendix for details and further results of patterns in unique vs. shared DEGs). We also found that *cis*-acting changes generally have larger effect sizes, as measured by absolute $\log_2$ fold change, consistent with previous studies in house mice (Figs 4C, 4D, S4C, and S4D) [16]. The difference in effect size between *cis*-only and *trans*-only was significant in both SARA and EDMA in BAT, but only in EDMA in liver (Welch two sample t-tests, Liver: SARA-MANA t = 1.55, df = 142.15, p = 0.124, EDMA-MANA t = 2.56, df = 408.85, p = 0.010, BAT: SARA-MANA t = 2.02, df = 183.35, p = 0.045, EDMA-MANA t = 3.12, df = 388.27, p = 0.002). Overall, we found that large-effect changes in *cis*- underlie much of expression evolution in this system.

## Different degrees of parallelism in *cis*- and *trans*-regulatory changes

Given that most expression evolution was due to changes in *cis*-, we hypothesized that a greater proportion of shared changes than expected by chance (as shown in Fig 2B for all DEGs)

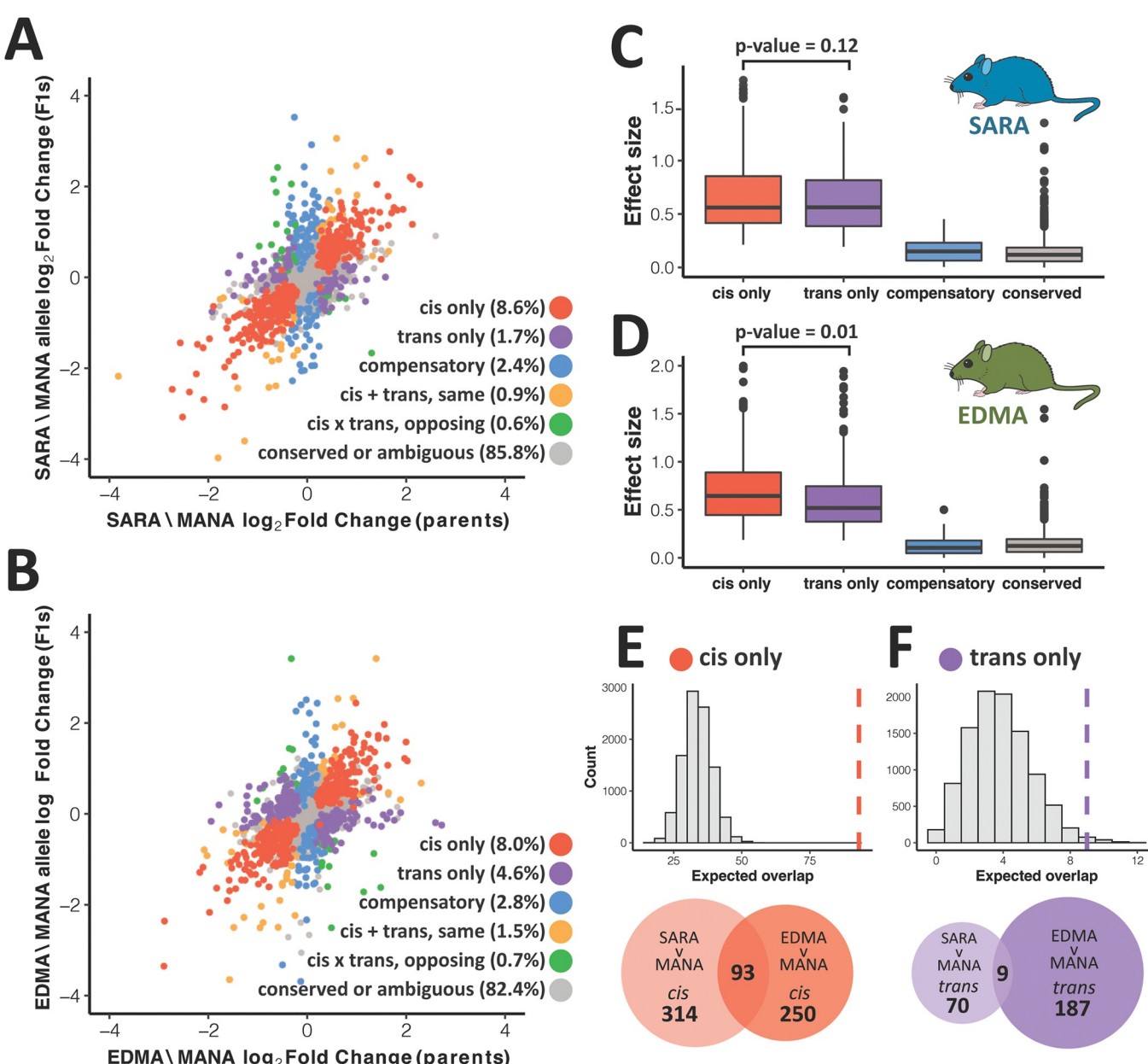

**Fig 4. *Cis-* and *trans-* gene regulatory evolution in liver.** A-B). Correlation between parental differential expression and differential expression between alleles in F1 hybrids for SARAvMANA (A) and EDMAvMANA (B) comparisons. Genes are colored based on regulatory mode (see methods for details). C-D). Effect size as measured as log$_2$ fold change for different regulatory classes in SARAvMANA (C) and EDMAvMANA (D) comparisons. P-values calculated using Welch two sample t-tests. E-F). Expected vs. observed overlap between SARAvMANA and EDMAvMANA *cis*-regulated (E) and *trans*-regulated (F) genes. Dashed line indicates observed overlap. Proportion of overlap shown in venn diagrams is significantly higher for *cis*-regulated genes as compared to *trans*-regulated genes (Chi-square test of independence, p < 0.0001).

would also be observed for DEGs controlled only in *cis*-. To test this hypothesis, we first compared the expected overlap between SARA and EDMA for DEGs in general to the observed overlap for genes defined as *cis*-regulated in both comparisons via permutation tests (see methods). The same test was done with *trans*-regulated genes. We focused on genes defined as "*cis*-only" and "*trans*-only", and not those that contained significant levels of both *cis*- and *trans*-regulation because the number of genes falling in those categories was small. Both *cis*- and

*trans*-regulated genes showed significant enrichment for parallel evolved DEGs (Figs 4E, 4F, S4E, and S4F, liver: p = ~0 for *cis*, p = 0.02 for *trans*, BAT: p = ~0 for *cis*, p = 0.002 for *trans*; observed values of overlap that are outside of the permuted expected distribution are reported as p = ~0).

To more directly compare the degree of parallelism in *cis*-regulated and *trans*-regulated genes, we analyzed the difference in proportion of shared genes via a chi-square test of independence. The average proportion of *cis*-regulated genes shared between comparisons was 0.25 for liver and 0.24 for BAT, while the average proportion shared for *trans*-regulated genes was 0.08 for liver and 0.09 for BAT (Figs 4E, 4F, S4E, and S4F). This difference in proportions is significant in both tissues (p < 0.0001 for liver and BAT). Additionally, for genes that were *cis*-regulated in both comparisons, we found that $\log_2$ fold changes between SARA and MANA and EDMA and MANA were highly correlated (S5 Fig, Spearman's r = 0.89 for liver, 0.96 for BAT). Because *cis*-regulated genes represent independent cases of expression divergence, whereas *trans*-regulated changes may influence the expression of many genes, this strong correlation in *cis*-regulated genes provides further evidence of significant parallelism among northern house mice. Together, these results suggest that while both *cis*- and *trans*-regulated genes are more likely than expected by chance to contribute to parallel expression divergence, *cis*-regulated genes are shared between parallel comparisons to a greater extent than *trans*-regulated genes.

## Tissue-specific genes are largely *cis*-regulated, exhibit high amounts of expression divergence, and are involved in parallel evolution

To investigate the importance of tissue-specificity in expression evolution, we integrated our liver and BAT RNA-seq data with a published adult mouse expression dataset [36] including seven tissues (brain, cerebellum, heart, kidney, liver, ovary, and testis), and calculated tissue specificity using the value Tau (see methods). Tau ranges from 0 to 1, with 0 corresponding to a gene evenly expressed across all tissues and 1 corresponding to a gene with expression restricted to one tissue [37]. For the SARA-MANA and EDMA-MANA comparisons, respectively, we identified 516 and 526 liver-specific genes, 161 and 161 BAT-specific genes, and 793 and 798 broadly expressed genes.

We then investigated the mode of regulatory control (i.e. *cis*- vs. *trans*-) for genes of differing tissue specificities using the regulatory categories determined via ASE in F1 hybrids. These analyses were done with a smaller set of genes for which we had gene regulatory data (see methods). Below we highlight the results for liver, but similar patterns were observed for BAT (S6 and S8 Figs). In liver, the vast majority of broadly expressed genes have conserved expression patterns between cold-adapted and warm-adapted mouse strains (SARA-MANA: 90.6%, EDMA-MANA: 87.4%). This proportion of conserved genes is greater than that seen across all genes (SARA-MANA: 80.9%, EDMA-MANA: 74.9%) (Figs 5A and S7A).

In contrast, for tissue-specific genes, only 46.7% and 35.8% were conserved for SARA-MANA and EDMA-MANA, respectively. Moreover, we found that a higher proportion of tissue-specific genes were regulated in *cis*- (SARA-MANA: 31.4%, EDMA-MANA: 33.9%) as compared to all genes (SARA-MANA: 11.6%, EDMA-MANA: 11.4%) and broadly expressed genes (SARA-MANA: 5.8%, EDMA-MANA: 4.6%) (Figs 5A and S7A). Further, using a permutation approach to estimate the expected number of DEGs, we found that tissue-specific genes are highly enriched for DEGs, suggesting that expression evolution is occurring in tissue-specific genes to a greater extent than in broadly expressed genes or in the transcriptome as a whole (Figs 5B, 5C, S7B, and S7C). Conversely, broadly expressed genes were under-enriched for DEGs, which is consistent with the large amount of conserved expression in this

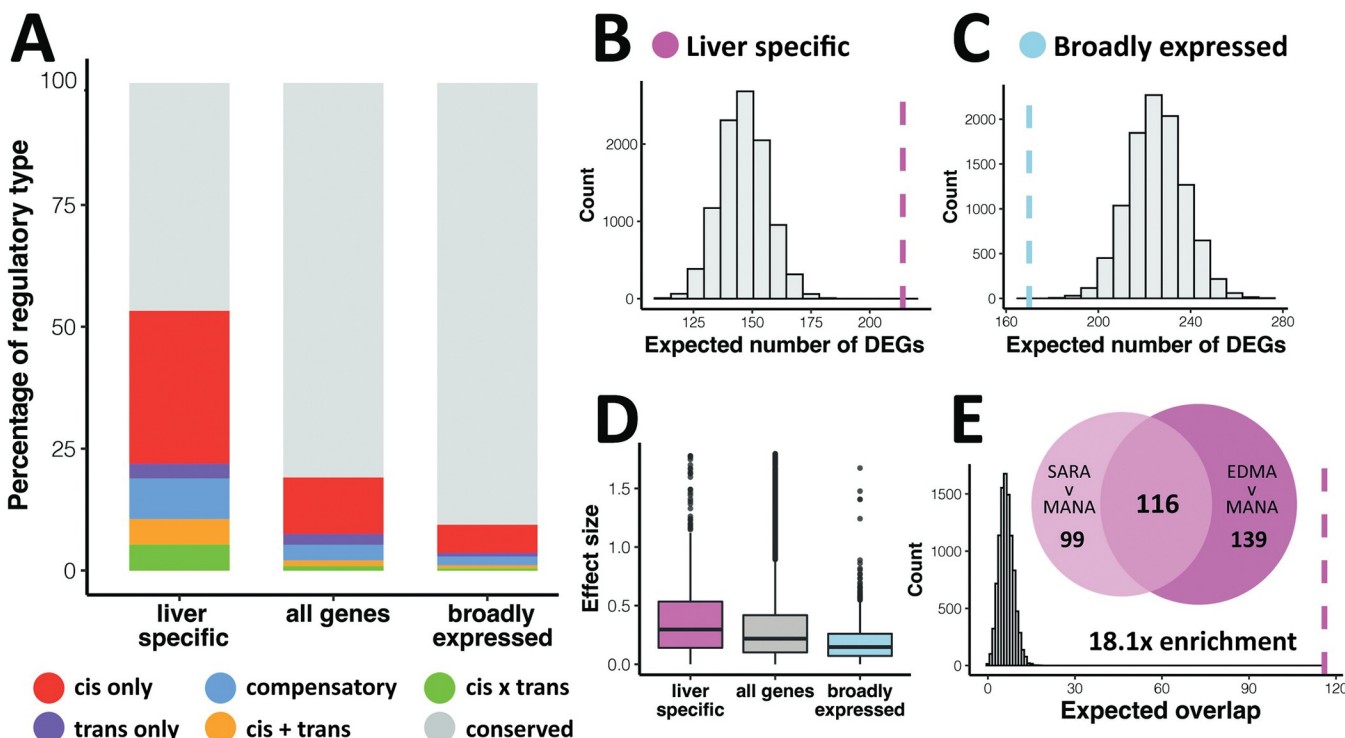

**Fig 5. Patterns of tissue-specific gene expression evolution between SARA and MANA mice in liver.** A). Percentage of each regulatory type in liver-specific, broadly expressed, and all genes. Genes of ambiguous regulatory type are not included. EDMAvMANA and BAT results can be found in S6–S8 Figs. B-C). Expected vs. observed number of significantly differentially expressed genes (DEGs) in liver-specific (B) and broadly expressed (C) genes. Dashed line indicates observed value. D). Effect size as measured as log$_2$ fold change for liver-specific, broadly expressed, and all genes. E). Overlap of liver-specific DEGs between SARAvMANA and EDMAvMANA comparisons (FDR < 5%) and expected vs. observed overlap between SARAvMANA and EDMAvMANA liver-specific DEGs. Dashed line indicates observed value.

gene set (Figs 5C and S7C). Finally, tissue-specific genes also showed larger effect sizes, as measured as absolute log$_2$ fold change (Figs 5D and S7D).

Given the high parallelism in *cis*-regulated genes, we hypothesized that differentially expressed, tissue-specific genes would be more likely to be involved in parallel evolution as compared to differentially expressed genes as a whole. Using permutation tests, we found that tissue-specific DEGs are more likely to be shared between SARA and EDMA than DEGs more generally for both liver and BAT (Figs 5E and S6E, liver-specific: p = ~0, BAT-specific: p = ~0).

## Signatures of selection to northern environments

While gene expression differences speak to patterns of regulatory evolution, they do not directly address whether these differences are adaptive. To link expression variation in lab mice to selection in natural populations, we used a normalized version of the population branch statistic (PBSn1), which identifies significant allele frequency shifts in a focal population in relation to two outgroup populations [38,39]. For focal populations representing MANA and EDMA lab mice, we utilized published whole exome data from Manaus, Brazil (MAN) and Edmonton, Alberta, Canada (EDM) [25,40]. For SARA we used published exome data collected near the border of New Hampshire and Vermont (NH/VT) [22], which is geographically close to Saratoga Springs, NY. The NH/VT population shows comparable patterns of similarity to SARA as MAN and EDM show to MANA and EDMA, respectively (see S1 Appendix and S13 Fig for details). We used published whole genome data from Iran and

France to serve as outgroup comparisons [41]. Iran and France are both ancestral populations to house mice in North and South America, making them suitable populations to serve as outgroups [20,25,42].

We first sought to identify selection occurring in the three focal groups, and thus performed three separate PBSn1 tests with NH/VT, EDM, and MAN as the focal population and France and Iran as outgroup populations. We calculated PBSn1 over 99,513, 109,113, and 106,141 non-overlapping 5-SNP blocks for the NH/VT, EDM, and MAN tests respectively, and we used BEDTools [43] to identify the closest gene to each SNP block. To connect signals of selection to differential expression, we looked for enrichment of genes showing significant allele specific expression (ASE), which is indicative of *cis*-acting regulation, among the top 1% of PBSn1 outliers. Using permutation tests (see methods), we found that ASE genes in either liver or BAT were significantly enriched among genes showing signals of selection in NH/VT and EDM, but not in MAN (S9 Fig, NH/VT: $p = 0.021$, EDM: $p = 2e-04$, MAN: SARAvMANA sig ASE, $p = 0.170$; EDMAvMANA sig ASE, $p = 0.593$). We saw the same pattern of significant enrichment only in the two northern populations in another set of PBSn1 tests that directly compared EDM to MAN and NH/VT to MAN using Iran as an outgroup (S1 Table). These results provide evidence of adaptive expression evolution in the two northern populations, but no evidence for adaptive expression evolution in the southern population, consistent with previous studies [44].

Given the lack of enrichment seen for significant ASE genes in MAN tests, we chose to focus solely on the two northern populations. We next sought to address the degree of parallelism seen in selection acting on northern populations. An important consideration is that EDM and NH/VT mice are more closely related to each other than they are to mice from Europe and the Middle East [25]. Therefore, shared outliers in the two PBSn1 tests using mice from France and Iran as outgroups can either be due to (1) shared ancestry between EDM and NH/VT or (2) true instances of parallel adaptation to a northern environment. To distinguish between these two possibilities, we performed a second (nested) set of PBSn1 tests focused on identifying outliers in NH/VT and EDM that are not associated with shared ancestry by using Arizona (AZ) and Iran as outgroups for EDM, and by using Florida (FL) and Iran as outgroups for NH/VT. NH/VT and FL mice form a clade, and EDM and AZ mice form a clade [25]. Therefore, shared outliers identified through this second set of tests reflect parallel adaptation to northern environments, as opposed to evolution occurring in the common ancestor of house mice that were introduced into North America.

The percentage of outliers retained between the EDM-France-Iran test and the EDM-AZ-Iran test (or the NH/VT-France-Iran test and the NH/VT-FL-Iran test) gives an estimate of the amount of shared PBSn1 outliers in the first test that are due to shared ancestry vs. true repeated instances of adaptation to a similar environment. Using this nested approach, we found that approximately 50% of outlier genes identified in the first tests were also identified in the nested tests (S10 Fig), suggesting that half of the outlier genes identified in the first tests were due to shared ancestry, while half represent instances of parallel adaptation.

## Enrichment of parallelism and *cis*-regulation among genes under positive selection

To identify selection acting on differential expression in cold-adapted house mice, we focused on the second set of PBSn1 tests above using Iran and AZ or FL as outgroups. For these tests we calculated PBSn1 over 82,809 and 100,602 non-overlapping 5-SNP blocks for NH/VT and EDM, respectively, and we used BEDTools [43] to identify the closest gene to each SNP-block (Fig 6A). Mirroring the extensive parallelism identified in the expression data (e.g. Figs 2B and

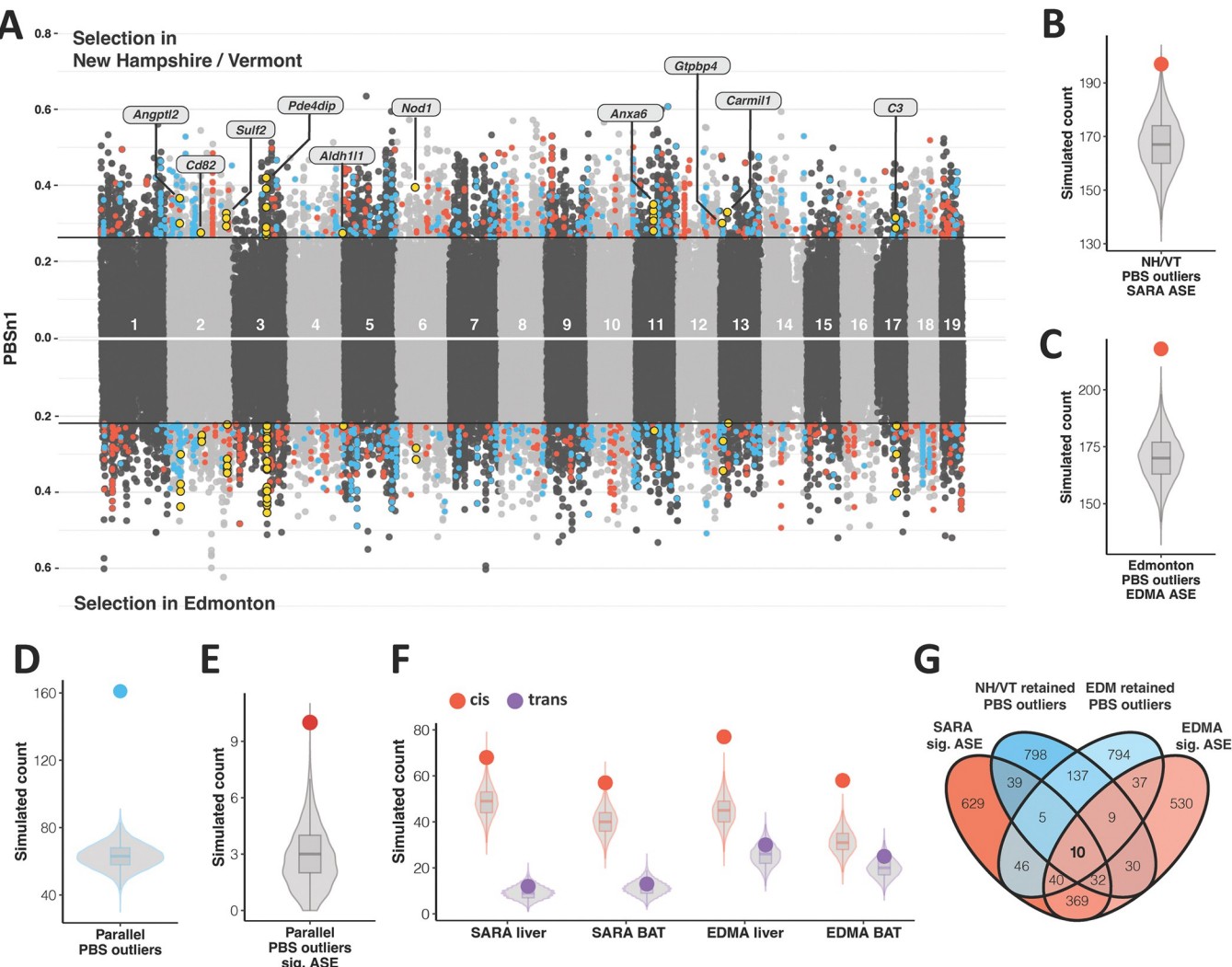

**Fig 6. Natural selection on *cis*-regulated genes and parallel expression divergence.** A). PBSn1 values for selection scans with New Hampshire/Vermont (NH/VT) (top) and Edmonton (EDM) (bottom) as the focal population. Line indicates the top 5% of 5-SNP blocks. Red points are PBSn1 outliers that overlap with a gene showing significant ASE in either liver or BAT. Blue points are parallel PBSn1 outliers (outliers that are retained in the nested tests and present in both the NH/VT and EDM tests). Yellow points indicate SNPs associated with the 10 genes that show significant ASE and are PBSn1 outliers in both comparisons. B-C). Expected vs. observed overlaps between PBSn1 outliers and significant ASE genes in either liver or BAT for NH/VT (B) and EDM (C) separately. Colored dot indicates observed value. D). Expected vs. observed overlap between NH/VT and EDM PBSn1 outliers that are retained in the nested tests. Blue dot indicates observed value. E). Expected vs. observed overlap between SARA and EDMA for independently evolved significant ASE genes (significant ASE genes that overlap a retained PBSn1 outlier in liver or BAT). F). Expected vs. observed overlaps between PBSn1 outliers and *cis*-only (red) or *trans*-only (purple) regulated genes. G).Venn diagram of PBSn1 outliers and significant ASE genes in either tissue for both comparisons.

S2B), we found that NH/VT and EDM PBSn1 outliers (top 5%) retained in the nested tests were more likely to be shared than expected by chance (Fig 6D, p = ~0). Second, we hypothesized that there would be significant overlap between PBSn1 outliers and genes showing significant ASE because we found *cis*-regulation to largely underlie expression divergence, and because *cis*-regulatory elements have long been hypothesized to be involved in adaptive evolution [1,7]. For both the SARA and EDMA comparisons, we found that significant ASE genes in either liver or BAT overlapped PBSn1 outliers more than would be expected by chance (Fig 6B–6C, p = 0.0025 for SARA, p = ~0 for EDMA). Interestingly, we also found significant overlap in *cis*-only regulated genes (p < 0.01 for both tissues in each comparison), while there was

not significant overlap of PBSn1 outliers and *trans*-only regulated genes (Fig 6F, p > 0.05 for both tissues in each comparison), which agrees well with *trans*-regulatory elements being unlinked from the genes they influence.

To address whether the shared variation seen among *cis*-regulated genes in the expression data (Figs 4E and S4E) was due to parallel evolution or shared ancestry among New York and Alberta mice, we performed another test for parallelism between SARA and EDMA using only significant ASE genes that overlap an outlier gene in the nested PBSn1 tests. Similar to the original tests for parallelism, we found significant parallelism between SARA and EDMA for significant ASE genes in either liver or BAT that are also under selection in wild populations (Fig 6E, p = 1e-04). We used an outlier cutoff of 5% in the nested PSBn1 tests to increase our power to detect relevant overlapping genes, however all significant enrichments were maintained at a cutoff of 1% (p < 0.05). These results provide strong evidence of parallel adaptive expression evolution.

To identify specific genes that may be directly involved in parallel adaptation to northern environments, we looked for overlap among genes that show significant ASE in both SARA and EDMA, and are under selection in both comparisons (Fig 6G). Of this set of 10 genes, many are implicated in body size variation (*Sulf2*, *Anxa6*, *Angptl2*) and immune functioning (*Nod1*, *C3*, *Cd82*) and the others are involved in activity level variation (*Pde4dip*), and coat abnormalities (*Gtpbp4*). Overall, we found significant parallelism in genes evolving under positive selection in cold-adapted house mice, that these genes are often associated with *cis*-regulatory elements, and that parallel, *cis*-regulated genes are involved in phenotypes such as body size differences and immune functioning.

## Discussion

Studying gene expression patterns in populations that have recently adapted to differing climates allowed us to characterize adaptive evolution during the earliest stages of divergence. Using allele-specific expression in two cold-adapted strains of mice in two tissues involved in metabolic functioning and thermogenesis, we sought to understand patterns of *cis*- and *trans*-regulation and tissue specificity during parallel adaptation to similar environmental conditions. SARA and EDMA mice have evolved both unique and shared gene expression changes compared to MANA mice, however the proportion of shared changes was greater than expected by chance. Further, expression changes at genes associated with increased body size were highly similar between SARA and EDMA mice, suggesting that New York and Alberta populations of house mice evolved the shared phenotype of increased body size through some shared molecular mechanisms. We found that tissue-specific and *cis*-acting changes formed the basis of most gene expression divergence between cold-adapted and warm-adapted strains of mice, and that these changes were more likely to contribute to parallel expression evolution. Lastly, comparison between the expression data and scans for selection in wild populations revealed significant parallelism in *cis*-regulated genes under positive selection in cold-adapted house mice. Together, these results provide strong evidence of adaptive expression evolution and lend insight into the nature of gene regulatory changes during rapid parallel adaptation to cold environments.

### Parallel gene expression and genomic changes underlie adaptation to cold environments

Repeated phenotypic changes in response to similar selective pressures provides an opportunity to study the repeatability of evolution at the molecular level. Additionally, repeated phenotypic change suggests that evolution is likely adaptive, and not a result of genetic drift. New

York and Alberta house mice have independently evolved larger body sizes and shortened extremities while adapting to a cold environment. These phenotypes are highly polygenic, and thus one might expect that at the genic level, the degree of parallelism involving these traits would be low [45]. In contrast to this expectation, we found expression evolution to be highly similar in EDMA and SARA mice in both liver and BAT, with approximately half of all differentially expressed genes being shared between comparisons.

Importantly, mice from New York and Alberta are more closely related to each other than they are to mice from Manaus (S11 Fig). Thus, some of the shared expression differences undoubtedly reflect shared ancestry, while some reflect true instances of parallel evolution. While we have no formal means for distinguishing between instances of parallel evolution and shared ancestry in the gene expression data directly, several of the expression analyses are consistent with the notion that a substantial fraction of the shared expression differences reflect parallel evolution. For example, the correlation between SARA-MANA expression differences and EDMA-MANA expression differences is greatest for genes involved in increased body size, a trait that is known to have evolved independently in New York and Alberta mice [25] (Fig 3). Additionally, we found significant overlap between *cis*-regulated genes and genes under selection in the nested PBSn1 test, which was designed to identify independently evolved differences in the two cold-adapted populations (Fig 6B–6C). Lastly, we found that the signal of parallelism among *cis*-regulated genes was maintained when using significant ASE genes that overlapped a PBSn1 outlier identified in the nested test (Fig 6E), suggesting that *cis*-regulated genes are under independent selection in both cold-adapted populations of house mice.

We found no enrichment of *cis*-regulatory divergence in genes under selection in Manaus house mice, further suggesting that expression differences identified here are more likely to reflect adaptation in the two northern populations. This lack of *cis*-regulatory enrichment in mice from Manaus was also found in Ballinger et al. [44]. The ancestral range of *M .m. domesticus* is in the Middle East and areas surrounding the Mediterranean Sea. These subtropical regions have relatively benign climates compared to those in Saratoga Springs, New York or Edmonton, Alberta. Thus, the selective pressure in these northern populations may have been much greater than those in the Manaus population. The substantial parallelism between cold-adapted house mice is consistent with selection on shared standing genetic variation, which is expected to increase the degree of similarity seen between independent cases of adaptation [45].

### *Cis*-regulated and tissue-specific changes dominate parallel expression evolution

We found evidence for a predominance of *cis*-regulated changes governing gene expression evolution in two instances of adaptation to northern environments. *Cis*-changes are generally thought to be less pleiotropic by allowing for tissue- and context-specific gene expression regulation [1,5,8]. Consistent with this, we also found that tissue-specific differentially expressed genes were largely *cis*-regulated and frequently contributed to expression evolution.

These results stand in contrast to the commonly reported finding that intraspecific gene expression variation is largely governed in *trans*-, while interspecific variation is largely governed in *cis*- [5,8–11]. However, the proportion of *cis*-regulatory variation that is discovered within species likely depends on many factors that differ among studies. First, studies that have found a predominance of *trans*-changes governing intraspecific variation have often focused on comparisons among individuals within a population such as inbred lab strains of *Drosophila* [9] or *Saccharomyces cerevisiae* strains derived from the same isolate [10,46]. In

contrast, in intraspecific comparisons where the groups have experienced directional selection, such as in freshwater vs. marine sticklebacks [19] or cold-adapted vs. warm-adapted *D. melanogaster* [47], *cis*-changes have sometimes been found to govern the majority of expression variation. These contrasting patterns can be explained if expression variation under stabilizing selection is largely governed in *trans*- while expression variation under positive directional selection is largely governed in *cis*-. Therefore, it seems important to consider whether the groups being compared are under divergent selection when interpreting the relative amounts of *cis*- and *trans*-regulation observed in intraspecific comparisons.

Second, the relative amount of *cis*- and *trans*-regulation that is discovered may depend on whether expression is studied in a specific tissue or in a whole organism. A bias could arise in two ways. First, because *cis*-acting changes are often associated with tissue-specific gene expression, measuring RNA at the level of the entire organism, such as an entire fly [9,11], could dampen the signal of tissue specific *cis*-changes, therefore minimizing the level of *cis*-regulation detected. Second, changes in cellular composition (i.e. changes in the proportions of different kinds of cells) will appear as *trans*-changes when studying expression in whole organisms. For example, flies with bigger brains will express more brain-specific transcripts simply as a consequence of having more brain cells. Interestingly, in studies that have analyzed the head and body separately in *Drosophila*, *cis*-acting changes were found to predominate [17].

While there are contrasting trends represented across various divergence times and organisms, here we present a case of nascent divergence (within ~500 years) facilitated largely by *cis*-changes; this trend was similarly demonstrated by Ballinger et al. [44] in their comparison of New York and Brazil house mice. Together, these studies lend support to the importance of *cis*-regulatory elements even during very early instances of intraspecific differentiation.

Further supporting a large role for *cis*-acting and tissue-specific changes, we found these modes of regulation to predominate in both SARA and EDMA mice, consistent with the idea that less pleiotropic changes are more likely to contribute to parallel divergence. Recent studies on gene regulatory patterns during parallel evolution have come to contrasting conclusions, with two studies finding a greater degree of parallelism in *cis*-acting changes [19,47] and another study finding a greater degree of parallelism in *trans*-acting changes [18]. Considerable previous theoretical and empirical work has suggested that less pleiotropic loci are more likely to contribute to parallel and convergent evolution [7,48–51]. However, some recent work in locally adapted stickleback populations has suggested that genes with intermediate levels of pleiotropy (identified through mapping studies and expression networks) may be more likely to underlie parallel adaptive evolution [52]. Genes with intermediate degrees of pleiotropy may be favored in situations where selection is acting on standing variation repeatedly in opposing directions. The distinction between *cis*- and *trans*- changes in our study does not allow us to assess "degrees" of pleiotropy, but future studies using eQTL mapping or gene co-expression networks, which can act as a proxy for pleiotropy, could be used to ask whether genes with intermediate levels of connectedness are most likely to underlie parallel evolution.

## Identifying candidate genes for complex traits

Uncovering the genetic basis of adaptation for complex traits is challenging because these phenotypes are controlled by a large number of genes of mostly small effect. Here we combined multiple experimental approaches including differential gene expression, scans for selection and the use of parallel instances of adaptation, that, in aggregate, provide more information on the genetic basis of adaptive, polygenic traits than any one method alone. Through this approach we were able to identify several genes that are differentially expressed, controlled in

*cis-*, and under selection in both northern populations. These genes are related to known phenotypic differences in northern and southern house mice including body size (*Sulf2*, *Anxa6*, *Angptl2*), immunity (*Nod1*, *C3*, *Cd82*), activity level (*Pde4dip*), with northern mice being more active [22,25], and coat differences (*Gtpbp4*). Interestingly, both the selection scans and gene ontology analyses (Figs 2D, 2E, S2D, and S2E) revealed a role for immune related genes during adaptation to northern environments. This result agrees with previous work showing novel pathogens as a strong selective force on organisms invading new environments, often shaping the response at the genomic level [53]. The use of multiple experimental approaches to identify potential candidate genes represents an initial step in finding the genes underlying adaptive evolution. These candidate genes can be further studied using methods that directly link phenotype to genotype through functional tests of specific mutations.

Our results show that selection is acting on *cis-*regulatory elements in natural populations to modulate gene expression levels that are then maintained in animals reared under common lab conditions. This work provides a detailed account of the gene regulatory changes underlying repeated instances of adaptation to cold environments and the degree of similarity between these cases of repeated adaptation. Ultimately, our findings underscore the importance of *cis-*regulated and tissue-specific changes in facilitating rapid adaptive evolution in organisms facing novel environmental conditions.

## Materials and methods

### Ethics statement

This work was conducted in accordance with the University of California, Berkeley Institutional Animal Care and Use Committee (AUP-2017-08-10248). Euthanasia was performed under approval of the relevant IACUC at the University of California, Berkeley, and included the humane use of isoflurane and cervical dislocation by trained personnel.

### Animal husbandry and mice used for this study

All mice were housed under a 12:12 light cycle at 70-72˚F and fed standard rodent food *ab libitum*. Wild derived, inbred strains of house mice used in this study were originally collected from 43˚N in Saratoga Springs, New York (SARA), 54˚N in Edmonton, Alberta, Canada (EDMA), and at 3˚S in Manaus, Brazil (MANA). Information on the establishment of these inbred strains is described in Phifer-Rixey et al. [22] and Ferris et al. [25]. In brief, live mice were collected from New York, Alberta, and Manaus, brought back to the lab, and unrelated individuals were mated to create the N1 generation. Inbred strains were generated via subsequent sib-sib mating. SARA, EDMA, and MANA mice in this study were past generation 10 of sib-sib mating and considered fully inbred.

We crossed SARA dams to MANA sires and EDMA dams to MANA sires to generate SARA x MANA and EDMA x MANA F1 hybrids, respectively. All mice were housed singly from weaning (3 weeks) until adulthood. The SARA, MANA, and SARA x MANA male F1 hybrids were previously generated in Ballinger et al. [44].

### Phenotyping, sample preparation and RNA-sequencing

At 11 weeks of age, six males of each group [SARA, EDMA, MANA, (SARAxMANA)F1, and (EDMAxMANA)F1] were sacrificed. Body weight, total body length, tail length, ear length, and hind foot length were measured, and liver and brown adipose tissue (BAT) were collected for RNA-sequencing. Further information for all mice used in this study can be found in S1 File. Liver and BAT were immediately placed in RNAlater, kept at 4˚C for 24 hours and then

stored at -80˚C until RNA-extraction. We extracted RNA using the Qiagen RNeasy minikit following the manufacturer's instructions. We validated RNA integrity using an Aligent Bioanalyzer and only RNA with RIN > 8 was used for library preparation. We generated cDNA libraries using the KAPA Stranded mRNA-Seq kit starting with 2ug of total RNA measured using Nanodrop, and uniquely indexed libraries using dual indexes from Illumina. We assessed library size and quality using an Aligent Bioanalyzer with a desired insert size of 300bp. Tissue-specific libraries were then pooled and sequenced across two lanes of Illumina 150bp paired-end NovaSeq, one using an S4 flow cell, and one using an S1 flow cell at the Vincent J. Coates Genomics Sequencing Center at UC Berkeley.

## Gene expression quantification and analysis

All analyses for the liver and BAT were performed identically, but separately unless otherwise noted. Resulting RNA-Seq reads were cleaned and trimmed with FastP [54] (v.0.19.6, parameters = –n_base_limit 5, –qualified_quality_phred 15, –unqualified_percent_limit 30, –detect_adapter_for_pe, –cut_window_size 4, –cut_mean_quality 15, –length_required 25), and then mapped to the mouse reference genome (GRCm38/mm10) with STAR [55] (v2.7.7a, parameters = –outFilterMultimapNmax 1, –outFilterMismatchNmax 3). After alignment, reads were counted using HTseq [56], and count files from each sample were merged using the python script merge_tables.py (Dave Wheeler, github link). This merged table was the input for downstream differential expression analyses.

Differential expression between mouse strains was quantified using DESeq2 in R [33], and only genes with a mean greater than 10 reads across samples were retained for analysis. This resulted in a set of 14,514 and 14,703 genes that were expressed in our liver and BAT datasets, respectively. The contrast parameter in the result command was used to isolate SARA vs MANA and EDMA vs MANA comparisons, and differential expression between strains was determined using Wald tests followed by a Benjamini-Hochberg correction for multiple testing with a false discovery rate (FDR) of 5%. DESeq2 was also used to generate principal component plots (Figs 1B and S1).

To confirm that the correlation in $\log_2$ fold change observed between the SARA vs MANA comparison and the EDMA vs MANA comparison (Figs 2C and S2C) was not a spurious correlation due to both variables having the same denominator, we re-quantified differential expression between the mouse strains as described above using different subsets of MANA mice in each comparison (ie. EDMA vs MANA samples 1-3 and SARA vs MANA samples 4-6). Correlations were maintained (S3 Fig), suggesting parallel changes.

## Gene set and gene ontology analyses

To test for increased gene expression parallelism associated with a parallel evolved phenotype, we compiled a list of genes linked to increased body size using the Mammalian Phenotype Browser on the Mouse Genome Informatics database (MP ID: 0001264). This returned a list of 412 genes to be used in downstream analyses (S2 File). We conducted gene ontology enrichment analyses separately for SARA and EDMA comparisons on differentially expressed genes at an FDR < 5% and a $\log_2$ fold change greater than ± 1 using PANTHER [34].

## Variant calling from transcriptomic data

We called single nucleotide polymorphisms (SNPs) using the Genome Analysis Tool Kit (GATK) [57] on our RNA-seq data following the RNA-seq best practices for germline short variant discovery. Reads were cleaned, trimmed, and aligned as described in the section above. We prepared BAM files for variant calling using Picard tools MarkDuplicates and

AddOrReplaceReadGroups, and by running SplitNCigarReads. We then ran HaplotypeCaller on each individual separately passing both the liver and BAT RNA-seq libraries as inputs. Separate gVCF files were combined using CombineGVCFs and genotypes were called on all individuals together using GenotypeGVCFs. We selected just SNPs using SelectVariants and filtered for high quality variants using the command VariantFiltration (parameters = QD >2, FS < 60, MQ > 40, MQRankSum > -12.5, ReadPosRankSum > -8).

## Allele specific expression analysis

To assign parental alleles for reads in F1 hybrids, we first had to identify fixed and different SNPs between the parent groups for each comparison. Starting from the high quality variants described above, we generated separate MANA, EDMA, and SARA VCF files using BCFtools and filtered for SNPs present in at least 85% of individuals using VCFtools [58]. We then created two VCF files for each parent group: one that contained SNPs fixed for the non-reference allele, and one that contained SNPs fixed for the reference allele. We then merged the resulting VCF files to generate a file for each comparison that only contained fixed and different SNPs between parent groups using vcf-merge. For example, for the EDMA vs MANA comparison, fixed non-reference allele EDMA SNPs were merged with fixed reference allele MANA SNPs, and vice versa for reference EDMA SNPs and non-reference MANA SNPs. These two resulting files were combined using BCFtools concat to generate a VCF file of only fixed and different SNPs between EDMA and MANA mice.

To differentiate between alleles in F1 hybrids, we called variants for F1s as described above, and then filtered for only heterozygous calls using BCFtools. To generate a set of ASE informative SNPs, we retained only variants present in each F1 individual and present in the fixed and different parent group VCF files. This resulted in a set of 30,000 ASE informative SNPs for the SARA vs MANA comparison and 25,229 SNPs for the EDMA vs MANA comparison. We phased the resulting VCF files for each comparison to indicate whether MANA or SARA/ EDMA contained the reference or alternate allele using custom scripts. We then generated two separate pools of reads for each F1 hybrid (one of MANA reads and one of SARA/EDMA reads) by aligning the F1 reads using STAR with WASP [59], specifying the ASE informative SNPs as the variant file. WASP adds a SAMtag to each read indicating its phase status in the provided VCF, here corresponding to the parental group to which the read maps. We selected for the different SAMtags using grep, generating two separate BAM files for each individual corresponding to reads of the different parental groups, which were subsequently counted using HTSeq count as described above. Reads from F1 hybrids equally mapped to both the MANA and SARA/EDMA allele, suggesting that there was no mapping bias toward either mouse strain (S12 Fig).

## *Cis*- and *trans*-regulatory assignment

We determined the relative amount of *cis*- and *trans*-regulated genes similarly to McManus et al. [60]. We performed three hierarchical statistical tests for differential expression using DESeq2 in R [33] only retaining genes with a mean count greater than 10 reads across all samples. Ultimately, we were able to test for ASE in 4,811 genes in liver and 4,502 genes in BAT for the SARA vs MANA comparison and 4,320 genes in liver and 3,864 genes in BAT for the EDMA vs MANA comparison. Genes able to be tested for ASE were those with a fixed and different SNP in the parents that was heterozygous in the F1s.

Three separate tests were used to analyze 1) differential expression between parental samples via Wald tests (P), 2) differential expression between the separate alleles of the F1 hybrids via Wald tests (H), and 3) the presence of *trans*- effects (T), which was determined using a

likelihood ratio test comparing the ratios of mouse strain expression difference between the parents and between the alleles of an F1 hybrid. A significant difference between ratios indicates that the expression difference between alleles in an F1 does not recapitulate the expression difference between the parents, pointing to the presence of *trans*-regulation. Genes were categorized into different regulatory modes based on their significance at an FDR of 5% in the P, H, and T statistical tests as described below.

- *cis-* only: significant in P and H, not significant in T

- *trans-* only: significant in P, not significant in H, significant in T

- *cis + trans*: significant in P, H and T. Expression divergence direction in P and H is the same.

- *cis x trans*: significant in P, H and T. Expression divergence direction in P and H is not the same.

- compensatory: significant in H, not significant in P, significant in T. No expression divergence due to compensation

- conserved: not significant in H, P, or T. No expression divergence

- ambiguous: Genes not encapsulated by above parameters.

Genes falling into each regulatory category (excluding ambiguous) for both comparisons and both tissues are listed in S3–S6 Files.

## Tissue specificity

To identify liver-specific and BAT-specific genes, we integrated our RNA-seq data with a published *Mus musculus* RNA-seq dataset from Cardoso-Moreira et al. [36] that analyzed expression from four mice (2 males and 2 females) in seven tissues: brain (forebrain/cerebrum), cerebellum (hindbrain/cerebellum), heart, kidney, ovary, testis, and liver. The Cardoso-Moreira et al. [36] dataset started as EdgeR generated reads per kilobase mapped (RPKM) values. To make our dataset comparable, we re-analyzed our HTSeq generated read counts (described in the "*Gene expression quantification and analysis*" section) as described in Cardoso-Moreira et al. [36]. Briefly, we used the TMM method in EdgeR [61] to normalize read counts, and generated RPKM values using the EdgeR rpkm function. We analyzed congruence between our dataset and the published dataset by comparing the separate liver RPKM values for each gene. Liver gene expression was highly correlated between the datasets (Pearson's R=0.75, $p < 2.2e^{-16}$), suggesting that our approach is reasonable.

We used tau to determine tissue specificity [37]. Tau captures the deviation in expression level for a gene in a particular tissue compared to its average FPKM across tissues. Tau ranges from 0-1, with 1 representing a gene only expressed in one tissue, and 0 representing a gene evenly expressed across all tissues [37]. Genes were defined as BAT- or liver-specific if tau > 0.85 [62] and if the highest FPKM was in BAT or liver, respectively. Genes were defined as broadly expressed if tau < 0.35. Because we had liver expression values both from the published dataset and our dataset, we calculated tau twice: once including our liver data, and once with the Cardoso-Moreira et al. [36] liver data. Only genes defined as liver-specific in both calculations were used in downstream analyses. Liver-specific, BAT-specific, and broadly expressed genes are listed in S7–S12 Files.

To investigate the relationship between tissue specificity and mode of regulation, we determined the regulatory class for each gene using the assignments generated in the "*cis- and trans-regulatory assignment*" section above. Genes of "ambiguous" regulatory class were not included. For the SARA vs MANA and EDMA vs MANA comparisons, respectively, we were

able to use 169 and 174 liver-specific genes, 20 and 12 BAT-specific genes, 277 and 237 broadly expressed genes in the liver dataset, and 279 and 242 broadly expressed genes in the BAT dataset that had both regulatory type data and tissue specificity data.

## Permutation tests of enrichment

To test for enrichment of parallelism in differentially expressed genes (DEGs), we performed permutation tests using the sample function in R without replacement with 10,000 iterations. Specifically, to test for parallelism in DEGs in liver, we randomly sampled 4,121 genes from the total of those expressed in the SARA vs MANA comparison (representing the number of DEGs between SARA vs MANA) and 4,936 genes from the total of those expressed in the EDMA vs MANA comparison. We then calculated the overlap between the randomly sampled gene sets and tabulated a p-value using the proportion of the simulated distribution that was outside the observed value. We calculated the degree of enrichment by dividing the observed overlap by the mean of the simulated overlap. Parallelism in *cis-* and *trans-*regulated genes was done in a similar manner by randomly sampling gene sets of the appropriate size for each comparison and each tissue from the total of those significantly differentially expressed (FDR < 0.05). Lastly, we tested for parallelism in differentially expressed tissue-specific genes analogously to that described above by randomly sampling from the set of genes significantly differentially expressed (FDR < 0.05).

We also tested for enrichment of expression divergence in tissue-specific and broadly expressed genes by comparing the observed vs. expected amount of DEGs in each gene set using a similar permutation approach as described above. This was carried out separately for the SARA vs MANA and EDMA vs MANA comparisons. More specifically, for liver-specific genes, we randomly sampled 230 genes (representing the number of liver-specific DEGs) from the total genes tested for tissue specificity, and then tabulated how many were significantly differentially expressed at an FDR < 5%. We repeated this analysis for BAT-specific and broadly expressed genes and calculated p-values and degree of enrichment as described in the paragraph above.

## PBS tests for selection

To identify loci under selection in wild populations of house mice, we used a normalized version of the population branch statistic (PBSn1), which identifies significant allele frequency shifts in a focal population in relation to two outgroup populations [38,39]. For natural populations comparable to expression data from EDMA, SARA, and MANA, respectively, we used previously published population level exome data from Edmonton, Alberta, Canada (EDM) [25], New Hampshire/Vermont (NH/VT), which is geographically close to Saratoga Springs, NY [22], and Manaus, Brazil (MAN) [40]. Details on exome capture design can be found in Phifer-Rixey et al. [22]. SNPs identified using exome probes are in exons, but a large proportion are also in adjacent intronic and inter-genic regions [22]. Because linkage disequilibrium (LD) in wild house populations extends for 10's of kb [63], SNPs in exome data are expected to be frequently linked to regulatory SNPs, making these data suitable for investigating regulatory evolution.

For outgroup populations we used genomic data from *M. m. domesticus* from France and Iran downloaded from https://wwwuser.gwdg.de/~evolbio/evolgen/wildmouse/ [41]. France and Iran were chosen as comparison populations as they are ancestral to house mouse populations in the Americas [20,25,42]. SNPs were called on the North and South American exomes together using GATK [57]. We ran HaplotypeCaller on each individual separately, and then combined gVCF files using CombineGVCFs. Genotypes were called on all individuals together

using GenotypeGVCFs. We selected just SNPs using SelectVariants and filtered for high quality variants using the command VariantFiltration (parameters = QD >2, FS < 60). We then selected for only the autosomes and combined the resulting VCF with the published VCF containing France and Iran mice. We filtered variants for those present in more that 80% of individuals and excluded those with a minor allele frequency < 5%, resulting in a set of 579,208 biallelic SNPs.

We used VCFtools [58] to calculate pairwise Weir and Cockerham Fst for each population comparison (see details below), and calculated PBSn1 in non-overlapping 5-SNP blocks as in Crawford et al. [39]. This SNP block approach results in a less diffuse signal, allowing us to identify regions of clear differentiation between the populations of interest. We then used BEDTools closest [43] to identify the closest gene in the mm10 Ensembl mouse genome annotation to each SNP block.

We calculated the parameters of the PBSn1 test shown below separately for EDM, NH/VT, and MAN as the focal population with France and Iran as the outgroup populations.

$$T = -\log(1 - F_{ST})$$

$$PBS = \frac{T^{pop1pop2} + T^{pop1pop3} - T^{pop2pop3}}{2}$$

$$PBS_{n1} = \frac{PBS_1}{1 + PBS_1 + PBS_2 + PBS_3}$$

In the PBS calculation, pop1 refers to the focal population, and in the PBSn1 calculation, PBS1 refers to the PBS value when EDM, NH/VT, or MAN is the focal population, while PBS2 and PBS3 refer to the PBS value when France and Iran are used as the focal populations, respectively. We also conducted tests analogous to those described above but directly comparing NH/VT and MAN with Iran as an outgroup, and EDM and MAN with Iran as an outgroup. These tests were done once with each northern group as the focal population and once with MAN as the focal population.

To address the confounding factor of shared ancestry of EDM and NH/VT, and to identify independent, parallel allele frequency changes in these two populations, we performed a "nested" PBSn1 test with two additional previously published North American population level exome samples, one from Arizona and one from Florida [22,25]. This second set of tests consisted of one with EDM as the focal population and Arizona and Iran as outgroups, and one with NH/VT as the focal population with Florida and Iran as outgroups. Because mice from EDM and NH/VT are more closely related to mice from Arizona and Florida, respectively, than they are to one another [25], PBSn1 hits identified in this second test that are shared between the EDM and NH/VT comparisons are most likely to be independent cases of parallel adaptation as opposed to shared ancestry. SNP calling and PBSn1 calculation was done similarly to that described for the first test. After all filtering steps, we retained a set of 667,423 biallelic SNPs. Ensembl IDs for genes overlapping outlier SNPs identified using BEDTools and for significant ASE and *cis*-regulated genes can be found in S13 File.

Tests for enrichment for parallel evolved PBSn1 outliers (FDR <0.05) between EDM and NH/VT and overlap between PBSn1 outliers and significant ASE genes, *cis*-only regulated genes and *trans*-only regulated genes were performed at the gene level analogously to that described in the "*Tests of enrichment*" section using the set of genes tested for ASE and in the relevant PBSn1 test as the background set.

## Phylogenetic tree construction

To describe the phylogenetic relationship between focal mouse groups, we used whole genome data from Iran and France from Harr et al. [41] as described above, the whole exomes of EDM, NH/VT, and MAN as described above. We constructed a maximum likelihood tree with RAxML (version 8.2, [64]) for these five populations using the GTRCAT model, specifying Iran as an outgroup. We visualized the best tree with iTOL [65] (S11 Fig).

## Supporting information

**S1 Appendix. Supplementary results of shared vs. unique DEG patterns and comparisons of exome and transcriptome variation.**
(DOCX)

**S1 Fig. Gene expression variation across tissues and samples.** A). Principal component analyses for all RNA-seq data separates samples first on tissue type, and second on original sampling locality. B). Principal component analyses for BAT RNA-seq data separates samples based on original sampling locality.
(TIF)

**S2 Fig. Parallel gene expression evolution in BAT.** A). Overlap of significantly differentially expressed genes (DEGs) between SARAvMANA and EDMAvMANA comparisons (FDR <5%). B). Expected vs. observed overlap between SARAvMANA and EDMAvMANA DEGs. C). Correlation between SARAvMANA $\log_2$ fold change and EDMAvMANA $\log_2$ fold change for all genes (black), SARAvMANA DEGs (blue) and EDMAvMANA DEGs (green). Dashed line represents reduced major axis regression for all genes. Correlation coefficients are Spearman's R. D-E). Gene ontology enrichment for SARAvMANA DEGs (D) and EDMAvMANA DEGs (E).
(TIF)

**S3 Fig. Parallel expression evolution with subsets of MANA data.** A-B). Correlation between SARAvMANA (MANA samples 4-6) $\log_2$ fold change and EDMAvMANA (MANA samples 1-3) $\log_2$ fold change in liver (A) and BAT (B). All expressed genes are in black, SARAvMANA DEGs are in blue and EDMAvMANA DEGs are in green. Dashed line represents reduced major axis regression for all genes. Correlation coefficients are Spearman's R. All significant correlations found using the full MANA data for each comparison are recapitulated.
(TIF)

**S4 Fig. *Cis*- and *trans*- gene regulatory evolution in BAT.** A-B). Correlation between parental differential expression and differential expression between alleles in F1 hybrids for SARAvMANA (A) and EDMAvMANA (B) comparisons. Genes are colored based on regulatory mode (see methods for details). C-D). Effect size as measured as $\log_2$ fold change for different regulatory classes in SARAvMANA (C) and EDMAvMANA (D) comparisons. P-values calculated using Welch two sample t-tests. E-F). Expected vs. observed overlap between SARAvMANA and EDMAvMANA *cis*-regulated (E) and *trans*-regulated (F) genes. Dashed line indicates observed overlap. Proportion of overlap shown in venn diagrams is significantly higher for *cis*-regulated genes as compared to *trans*-regulated genes (Chi-square test of independence, p < 0.0001).
(TIF)

**S5 Fig. Parallel expression evolution in *cis*-regulated genes.** Correlation between SARAvMANA $\log_2$ fold change and EDMAvMANA $\log_2$ fold change in liver (A) and BAT (B) for genes that are *cis*-regulated in both comparisons. Dashed line represents reduced major axis

regression. Correlation coefficients are Spearman's R.
(TIF)

**S6 Fig. Patterns of tissue-specific gene expression evolution between SARA and MANA mice in BAT.** A). Percentage of each regulatory type in BAT-specific, broadly expressed, and all genes. Genes of ambiguous regulatory type are not included. B-C). Expected vs. observed number of significantly differentially expressed genes (DEGs) in BAT-specific (B) and broadly expressed (C) genes. Dashed line indicates observed value. D). Effect size as measured as $\log_2$ fold change for BAT-specific, broadly expressed, and all genes. E). Overlap of BAT-specific DEGs between SARAvMANA and EDMAvMANA comparisons (FDR < 5%) and expected vs. observed overlap between SARAvMANA and EDMAvMANA BAT-specific DEGs. Dashed line indicates observed value.
(TIF)

**S7 Fig. Patterns of tissue-specific gene expression evolution between EDMA and MANA mice in liver.** A). Percentage of each regulatory type in liver-specific, broadly expressed, and all genes. Genes of ambiguous regulatory type are not included. B-C). Expected vs. observed number of significantly differentially expressed genes (DEGs) in liver-specific (B) and broadly expressed (C) genes. Dashed line indicates observed value. D). Effect size as measured as $\log_2$ fold change for liver-specific, broadly expressed, and all genes.
(TIF)

**S8 Fig. Patterns of tissue-specific gene expression evolution between EDMA and MANA mice in BAT.** A). Percentage of each regulatory type in BAT-specific, broadly expressed, and all genes. Genes of ambiguous regulatory type are not included. B-C). Expected vs. observed number of significantly differentially expressed genes (DEGs) in BAT-specific (B) and broadly expressed (C) genes. Dashed line indicates observed value. D). Effect size as measured as $\log_2$ fold change for BAT-specific, broadly expressed, and all genes.
(TIF)

**S9 Fig. Enrichment of *cis*-acting regulation genes among PBSn1 outliers in the three focal groups.** PBSn1 values and expected vs. observed overlaps between genes showing significant ASE and top 1% PBSn1 outliers in the NH/VT-France-Iran test (A) the EDM-France-Iran test (B) and the MAN-France-Iran test (C). Line in Manhattan plots indicates the top 1% cutoff, and colored dots in the permutation test distribution plots indicates the observed overlap.
(TIF)

**S10 Fig. Shared outlier genes in nested PBSn1 tests.** A-C). Overlap of the top 10% (A), 5% (B) and 1% (C) of outlier genes shared between the NH/VT-France-Iran and NH/VT-FL-Iran PBSn1 tests. D-F). Overlap of the top 10% (D), 5% (E) and 1% (F) of outlier genes shared between the EDM-France-Iran and EDM-AZ-Iran PBSn1 tests.
(TIF)

**S11 Fig. Relationship among focal populations.** Maximum likelihood tree of focal house mouse populations from the ancestral range and North and South America. The tree is rooted using eight samples from Iran.
(TIF)

**S12 Fig. Lack of mapping bias in F1 hybrids.** A-D). Ratio of F1 hybrid reads mapping to the MANA allele vs. all mapped reads is centered around 0.5 in each tissue for SARAxMANA (A-B) and EDMAxMANA (C-D) F1 hybrids, suggesting there is no mapping bias for reads

preferentially mapping to the allele of one parent.
(TIF)

**S13 Fig. Relationship between exome and transcriptome genetic variation.** A). PCA of shared SNPs among focal population exome (Manaus, Edmonton, and NH/VT) and transcriptome (MANA, EDMA, and SARA) samples. B). Midpoint rooted neighbor joining tree of exome and transcriptome samples.
(TIF)

**S1 Table. Direct comparison PBSn1 tests.** Enrichment of significant ASE genes in either liver or BAT among top 1% PBSn1 outliers in tests directly comparing NH/VT vs Manaus and Edmonton vs Manaus. Only PBSn1 outliers able to be tested for ASE and significant ASE genes represented in the relevant PBSn1 test are used. P-values represent percent of permuted distribution lying outside the observed overlap. Observed overlaps entirely outside the distributed are represented as p = ~0.
(DOCX)

**S1 File. Phenotypic data and metadata for mice used for RNA sequencing.** Unique identifiers, sample names, other metadata, and phenotypic measurements of mice used for RNA sequencing.
(XLSX)

**S2 File. Increased body size genes.** Ensembl IDs for increased body size genes identified using the Mammalian Phenotype Browser on the Mouse Genome Informatics database (MP ID: 0001264).
(XLSX)

**S3 File. SARAvMANA liver gene regulatory categories.** Genes (Ensembl IDs) falling into different regulatory categories. Log2 fold changes and adjusted p-values for DESeq2 tests on parental differential expression, F1 allele differential expression, and trans expression. Parental base mean also listed.
(XLSX)

**S4 File. SARAvMANA BAT gene regulatory categories.** Genes (Ensembl IDs) falling into different regulatory categories. Log2 fold changes and adjusted p-values for DESeq2 tests on parental differential expression, F1 allele differential expression, and trans expression. Parental base mean also listed.
(XLSX)

**S5 File. EDMAvMANA liver gene regulatory categories.** Genes (Ensembl IDs) falling into different regulatory categories. Log2 fold changes and adjusted p-values for DESeq2 tests on parental differential expression, F1 allele differential expression, and trans expression. Parental base mean also listed.
(XLSX)

**S6 File. EDMAvMANA BAT gene regulatory categories.** Genes (Ensembl IDs) falling into different regulatory categories. Log2 fold changes and adjusted p-values for DESeq2 tests on parental differential expression, F1 allele differential expression, and trans expression. Parental base mean also listed.
(XLSX)

**S7 File. Liver-specific genes in the SARAvMANA comparison.** Published RPKM means (prefix "pub"), liver and BAT RPKM means from this study (prefix "local"), and calculated tau

values using both published and locally generated liver mean expression values.
(XLSX)

**S8 File. BAT-specific genes in the SARAvMANA comparison.** Published RPKM means (prefix "pub"), liver and BAT RPKM means from this study (prefix "local"), and calculated tau values using both published and locally generated liver mean expression values.
(XLSX)

**S9 File. Broadly expressed genes in the SARAvMANA comparison.** Published RPKM means (prefix "pub"), liver and BAT RPKM means from this study (prefix "local"), and calculated tau values using both published and locally generated liver mean expression values.
(XLSX)

**S10 File. Liver-specific genes in the EDMAvMANA comparison.** Published RPKM means (prefix "pub"), liver and BAT RPKM means from this study (prefix "local"), and calculated tau values using both published and locally generated liver mean expression values.
(XLSX)

**S11 File. BAT-specific genes in the EDMAvMANA comparison.** Published RPKM means (prefix "pub"), liver and BAT RPKM means from this study (prefix "local"), and calculated tau values using both published and locally generated liver mean expression values.
(XLSX)

**S12 File. Broadly expressed genes in the EDMAvMANA comparison.** Published RPKM means (prefix "pub"), liver and BAT RPKM means from this study (prefix "local"), and calculated tau values using both published and locally generated liver mean expression values.
(XLSX)

**S13 File. Summary of genes overlapping PBSn1 outlier hits and *cis*-regulated genes.** Ensembl IDs for NH/VT and EDM PBSn1 outliers (FDR <0.05), parallel outliers, outliers overlapping significant ASE genes, and outliers overlapping *cis*-only regulated genes.
(XLSX)

## Acknowledgments

The authors are grateful for the help from many scientists: Katya Mack, Libby Beckman, and Yocelyn Gutierrez Guerrero for assistance and advice on computational analyses; Lydia Smith and the Evolutionary Genomics Lab for guidance during RNA library preparation; Clara Drew for feedback on statistical analyses; and members of the Nachman lab for helpful discussions over the duration of this project.

## Author Contributions

**Conceptualization:** Sylvia M. Durkin, Mallory A. Ballinger, Michael W. Nachman.

**Formal analysis:** Sylvia M. Durkin.

**Funding acquisition:** Michael W. Nachman.

**Investigation:** Sylvia M. Durkin, Mallory A. Ballinger.

**Supervision:** Michael W. Nachman.

**Visualization:** Sylvia M. Durkin.

**Writing – original draft:** Sylvia M. Durkin, Michael W. Nachman.

**Writing – review & editing:** Mallory A. Ballinger.

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
