## [Decision Letter · Decision Letter 0]

25 Sep 2023

Dear Dr Durkin,

Thank you very much for submitting your Research Article entitled 'Tissue-specific and cis-regulatory changes underlie parallel, adaptive gene expression evolution in house mice' to PLOS Genetics.

The manuscript was fully evaluated at the editorial level and by independent peer reviewers. The reviewers appreciated the attention to an important problem and enjoyed reading the manuscript, but raised some substantial concerns about the current manuscript. Based on the reviews, we will not be able to accept this version of the manuscript, but we would be willing to review a much-revised version. We cannot, of course, promise publication at that time. With regard to the main comment of Reviewer 2, I wonder if the transcription data, coupled with imputation, could be provide sufficient information on genetic variation to conduct a selection scan and possibly validate the results of the scan presented in the current manuscript.

If you decide to revise the manuscript for further consideration at PLOS Genetics, please aim to resubmit within the next 60 days, unless it will take extra time to address the concerns of the reviewers, in which case we would appreciate an expected resubmission date by email to plosgenetics@plos.org.

We are sorry that we cannot be more positive about your manuscript at this stage. Please do not hesitate to contact us if you have any concerns or questions.

Yours sincerely,

Anna Di Rienzo

Academic Editor

PLOS Genetics

Kelly Dyer

Section Editor

PLOS Genetics

Reviewer's Responses to Questions

**Comments to the Authors:**

Reviewer #1: The study of house mice colonizing and adapting to the Americas from their ancestral range about 500 years ago continues to fascinate. This manuscript focuses on understanding the genetics underlying cold adaptation. By integrating expression data from mice reared under common laboratory conditions, in addition to controlled crosses allowing the assignment of cis vs trans regulatory changes, and genome wide SNP data analyses, the authors make a rather convincing point that adaptation to cold habitats involved mostly the same molecular changes in two widely separated populations (Saratoga Spring, New York and Edmonton, Alberta). They show that this evolution involves expression changes that are more likely to be cis regulatory, more likely to be tissue specific and more likely to show a signature of selection at the genomic level than expected by chance, and affect phenotypes such as body size, immune function and skeletal morphology.

I don’t have very much to criticize with data analyses, presentation and conclusions of this paper. As the authors state, finding the genetic basis of environmental adaptation is still a difficult task, and these data further our understanding significantly. My only point would be that the story is somewhat incomplete with respect to three aspects (detailed further below).

1) Terminology of “parallelism”

To me, the word parallelism implies that there is no room for the possibility that cold adapted alleles could have spread from one transect to the other by gene flow. In the earlier Plos Genetics paper frequently referred to the authors state: “An alternative possibility is that adaptation to high latitudes occurred once and that beneficial alleles were carried by rare migrants between the

eastern and western transects. This seems less likely in light of the very recent colonization of house mice in the Americas and the well-supported phylogenetic relationships depicted in Figs 1B and S2”.

However, this possibility is completely ignored in the current paper. Long distance migration is common in house mice and it is now generally well known that just a trickle of gene flow is enough for adaptive introgression of alleles across populations, with many examples emerging. To me, it is still an open question to what degree the correlations observed in the data stem from

true parallel evolution in situ rather than through some form of admixture, or a combination of both. A correlation is all that has been found so far. The authors themselves state: “Tissuespecific and cis-regulatory changes underlie correlated adaptive gene expression evolution in house mice.” I feel that the word “correlation” or “correlated” would represent the current understanding of the system better than parallel or parallelism. Because the adaptive changes could result from introgression or independent selection from standing variation, I feel a better title would be to ignore these alternatives. The title therefore could read:

“Tissue-specific and cis-regulatory changes underlie adaptive gene expression evolution in house

mice.”

This does not take away from the value of the paper, as similarities in both transects still speak strongly for adaptation being the driving force for the phenotypic and expression changes observed. And all other conclusions remain the same.

2) Unique changes are not described

My second major point is somewhat related to the above and concerns the fact that in this paper no attempt is made to characterize the unique changes in more detail. Edmonton is 10 degrees further North with more extreme cold weather compared to Saratoga Spring. Unique changes are expected to contribute to adaptation to those differences in climate. I would be interested in knowing to what degree unique changes show similar characteristic than shared changes (i.e. are they also more likely to be cis and more likely to be tissue specific and more likely found in “increase growth genes” and more likely to show a signal of positive selection?). This information would give a somewhat more complete picture on just how similar both transects are. Of interest here is the fact that the Edmonton mice show a comparatively much larger proportion of expression changes in trans (Fig. 4E, 4F). In addition, there are actually more regulatory changes that are unique than shared in the Edmonton/Manaus comparison. I wonder if these results could be interpreted in such a way that adaptation to the more “benign” Saratoga Spring conditions are primarily in cis, but more extreme adaptations to even colder conditions

require not only more changes but also more dramatic changes in trans?

3) Selection pressures on southern habitats are not assessed.

Some of the expression differences observed in the comparisons between Manaus and the northern populations are expected to be due to adaptation of Manaus mice to warm habitats. The authors acknowledge this but make no attempt to estimate how much of the differences can be attributed to it, although early in the paper we are led to believe they will address the issue. I understand that this is a formidable challenge and as far as I can see can only be done indirectly, by looking at patterns of selection on the genomic level. So I am wondering why did the scan for selection not include the Manaus population? If the authors only intend to describe the genetic basis of cold adaptation, maybe it should be included in the title?

Minor points:

Figure S11: I wonder if the authors could show branch length in the tree. This would give a better idea about overall divergence between the populations.

Fig.2 lines should be reduced major axis or PCs or 95% ellipses not linear regression, as the slope of the line will depend on which data are plotted on the X or Y axis.

Fig 4: P-values calculated using paired t-tests: what is the deg freedom

Page 22, line 484: should read …. 70-72F or 21-22C

Reviewer #2: This paper examines parallel gene expression evolution in two replicate cold adapted populations of house mice. Using mice derived from two independent colonizations of cold environments (New York US, and Edmonton CA) the authors show that a large proportion of gene expression changes have evolved in parallel between the two cold-adapted populations, most of the parallel expression phenotypes are driven by cis-regulatory variants, and many of the parallel genes may bear signature of selection in cold environments (though there are caveats here - see below). In general, the paper focuses on an interesting and important set of questions, is exceptionally well-written, and the analyses seem technically sound and well-executed (for the most part). It’s an interesting paper, and I enjoyed reading it.

My only major critiques are associated with the genomic selection scans, which seemed a bit clunky. As a general critique, there is a lack of overlap between the populations used in the transcriptomic analyses and those used in the genome scans. While the experiments used to identify gene expression evolution were focused on mice from New York, Edmonton, and Brazil, the populations used in the genome scans, varied depending on the comparison. However, there was the only direct overlap for mice from Edmonton (and even here it is not clear if the same locality was sampled). Mice sampled from Vermont and New Hampshire served as a proxy for the New York mice, and there was no analysis of genetic variation in mice from warm-adapted Brazilian population. Since mice mice from New York and Brazil, were used to identify morphological and regulatory phenotypes associated with cold adaptation, it seems odd to exclude them from the analyses designed to detect genomic signatures of selection.

The reasons for this omission stem largely from the fact that genome-scans involved a reanalysis of previously published data - there were no population genomic data generated in this study (to my understanding anyway). While I understand and appreciate the value of reanalyzing existing datasets, in this case, the approach does not seem ideal. The genome scans relied on the population-branch statistic, which of course, requires a three-way comparison allele frequency variation. Because the transcriptomic analysis involved two cold-adapted populations and one warm-adapted population, it was not possible to identify loci that specifically bear signatures of cold-adaptation using a PBS analysis of those three focal populations, and the authors recognized this problem. Thus, by necessity, the authors needed to sample additional outgroups. It seems that the best design for these comparisons, however, would have involved a cold-adapted population (NY or ED), Brazil, and an appropriate warm-adapted outgroup, repeating the analysis for each focal cold adapted population. This design could replace the authors' 'nested analysis' and the outliers in these repeated PBS analysis could more cleanly be interpreted as loci under parallel selection in each of the cold-adapted populations and related back to the gene expression patterns.

The authors did something like this, but the outgroups varied by comparison, proxy samples were used for one cold-adapted population (i.e. NH/VT to represent NY), and the focal warm-adapted population was never examined. These issues make relating the results of the genome scans to the patterns gene expression evolution a bit murky. The authors do address some of these issues (e.g. lines 295-301), but much of that discussion focuses on justification of Eurasian mice as outgroups. This is an important consideration, but at least equally important is whether VT/NH is an appropriate proxy for NY. The sole justification is that these locations are “geographically close” (line 295). Is there any appreciable population structure between these two localities? Do VT/NH mice have similar body sizes and limb lengths as the NY mice? These details could and should be easily added to better justify the choice. Similarly, why weren’t genomic data from Brazilian mice included? Are they not publicly available, or is another, better-grounded reason?

Similarly, the data from both the Edmonton and VT/NH mice were exome data (line 296). This seems an odd choice for a paper aimed understanding regulatory evolution. How confident are the authors that this strategy captures variation in and selection on relevant cis-regulatory elements? Details on exome capture design, like whether and how much flanking/non-exon sequence was captured would be useful, but again, it begs the question of whether the publicly available data are well-suited to address the questions at hand.

All of these issues could have been avoided if population genomic data (e.g. low coverage whole genome data) had been gathered for the three focal populations here. That said, I am certainly sensitive to financial and other constraints that often preclude optimal study design.

Below are some more minor points for the authors to consider:

Line 267-268: If space constraints are not too stringent, it would be better to present the BAT results in the main text rather than relegating them to the supplement.

Lines 455-458: This is a fine suggestion, but I think it is important to remind the readers that co-expression networks (e.g. WGCNA) are correlation networks and not true interaction networks, and as such, are not a direct measure of the extent of pleiotropy.

Lines 460-481: This section seems a little premature/overstated since no direct phenotype-genotype connection has been made here. I do agree that this approaches such as those employed here are a good way to nominate candidate genes for further analysis though. Consider sharpening the focus a bit more, and make clearer that variation (neither expression nor allelic) has not been directly tied to adaptive phenotypes in this study. That distinction might not be clear to the casual reader in its current state.

**Have all data underlying the figures and results presented in the manuscript been provided?**

Reviewer #1: Yes

Reviewer #2: **No: **It seems as though the RNASeq have not yet been submitted to the NCBI SRA. The cover letter includes a place holder number for the BioProject ID.

PLOS authors have the option to publish the peer review history of their article (what does this mean?). If published, this will include your full peer review and any attached files.

Reviewer #1: No

Reviewer #2: No

---

## [Decision Letter · Decision Letter 1]

22 Jan 2024

Dear Dr Durkin,

We are pleased to inform you that your manuscript entitled "Tissue-specific and cis-regulatory changes underlie parallel, adaptive gene expression evolution in house mice" has been editorially accepted for publication in PLOS Genetics. Congratulations!

Yours sincerely,

Anna Di Rienzo

Academic Editor

PLOS Genetics

Kelly Dyer

Section Editor

PLOS Genetics

Comments from the reviewers (if applicable):

Reviewer's Responses to Questions

**Comments to the Authors:**

Reviewer #2: The authors have done a very nice job addressing my previous concerns with the earlier version of the manuscript. I have nothing more to add. This is nice contribution and I look forward to seeing it in print.

**Have all data underlying the figures and results presented in the manuscript been provided?**

Reviewer #2: Yes

PLOS authors have the option to publish the peer review history of their article (what does this mean?). If published, this will include your full peer review and any attached files.

Reviewer #2: No

**Data Deposition**

http://datadryad.org/submit?journalID=pgenetics&manu=PGENETICS-D-23-00864R1

**Press Queries**

---

## [Editor Report · Acceptance letter]

29 Jan 2024

PGENETICS-D-23-00864R1 

Tissue-specific and cis-regulatory changes underlie parallel, adaptive gene expression evolution in house mice 

Dear Dr Durkin, 

We are pleased to inform you that your manuscript entitled "Tissue-specific and cis-regulatory changes underlie parallel, adaptive gene expression evolution in house mice" has been formally accepted for publication in PLOS Genetics! Your manuscript is now with our production department and you will be notified of the publication date in due course.

With kind regards,

Anita Estes

PLOS Genetics

On behalf of:
